# DNA Methylation as a Biomarker for Monitoring Disease Outcome in Patients with Hypovitaminosis and Neurological Disorders

**DOI:** 10.3390/genes14020365

**Published:** 2023-01-30

**Authors:** Olaia Martínez-Iglesias, Vinogran Naidoo, Lola Corzo, Rocío Pego, Silvia Seoane, Susana Rodríguez, Margarita Alcaraz, Adriana Muñiz, Natalia Cacabelos, Ramón Cacabelos

**Affiliations:** EuroEspes Biomedical Research Center, International Center of Neuroscience and Genomic Medicine, 15165 Bergondo, Corunna, Spain

**Keywords:** DNA methylation, folic acid, vitamin B12, psychometric testing, biomarker

## Abstract

DNA methylation remains an under-recognized diagnostic biomarker for several diseases, including neurodegenerative disorders. In this study, we examined differences in global DNA methylation (5mC) levels in serum samples from patients during the initial- and the follow-up visits. Each patient underwent a blood analysis and neuropsychological assessments. The analysis of 5mC levels revealed two categories of patients; Group A who, during the follow-up, had increased 5mC levels, and Group B who had decreased 5mC levels. Patients with low Fe-, folate-, and vitamin B12- levels during the initial visit showed increased levels of 5mC after treatment when assessed during the follow-up. During the follow-up, 5mC levels in Group A patients increased after treatment for hypovitaminosis with the nutraceutical compounds Animon Complex and MineraXin Plus. 5mC levels were maintained during the follow-up in Group A patients treated for neurological disorders with the bioproducts AtreMorine and NeoBrainine. There was a positive correlation between 5mC levels and MMSE scores, and an inverse correlation between 5mC and ADAS-Cog scores. This expected correlation was observed in Group A patients only. Our study appears to indicate that 5mC has a diagnostic value as a biomarker across different pathologies.

## 1. Introduction

Epigenetics is the study of reversible heritable changes in gene expression that occur without modifications to the DNA sequence, and links the genome with the environment [1,2]. DNA methylation, the most common and best-characterized epigenetic mechanism, is a reversible process mediated by DNA methyltransferases (DNMTs) in which methyl groups from S-adenosyl-l-methionine (SAM) are added to cytosines in CpG nucleotides, converting them to 5-methylcytosines (5mC) [3]. This process modifies DNA accessibility and stability, thus regulating gene expression [4]. DNA methylation is usually a repressive mark [5] that attracts other silencing elements, such as methyl-CpG-binding proteins [6]. Targeting the epigenome is an optimal tool for drug discovery because of the reversibility of epigenetic modifications. Moreover, the significance of epigenetic biomarkers in aiding patient diagnosis and prognosis, and their responses to therapy, is becoming increasingly recognized.

DNA methylation correlates with certain neuropsychiatric and neurodegenerative disorders, such as schizophrenia, Rett syndrome, Alzheimer’s disease (AD), Parkinson’s disease (PD), and Huntington’s disease [7,8]. The roles of DNA methylation in neural development, plasticity and learning and memory have also been explored using DNMT inhibitors (e.g., 5-aza-2′-deoxyazacytidine, zebularine), DNMT1-knockout mice, and DNA methylation-related proteins (e.g., methyl-CpG-binding proteins) [7]. Global DNA methylation is reduced in the AD human glioblastoma H4-sw cell line [9] and is lower in brain and serum samples in triple-transgenic (3xTg)-AD mice than in wild-type mice [10]. DNMT3a expression is also reduced in the 3xTg-AD mouse brain compared to wild-type animals [10]. Treating mice with the DNMT inhibitors RG108 or 5-aza-2′-deoxycytidine enhance the behavioral effects of antidepressants and modulate stress-induced DNA methylation/demethylation [11].

DNA methylation is reduced in temporal neocortical cells in monozygotic twins discordant for AD [12] and in the hippocampus in patients with AD [13]. Global DNA methylation levels are also low in blood samples from patients with cerebrovascular and neurodegenerative disorders [10,14,15,16]. Given that epigenetic changes are reversible, DNA methylation may therefore be used as a treatment-outcome/response biomarker. Two possible causes of the aberrant regulation of DNA methylation in brain disorders are aging [17] and a deficiency in vitamin B [18]. DNA methylation-based age clocks are proposed as biomarkers of early disease risk and as predictors of life expectancy and mortality [19]. Folate and vitamins B6 and B12 are key contributors in biochemical reactions for the methylation of DNA, and vitamin B deficiency may block the DNA methylation metabolic cycle [18]. DNA methylation is also implicated in other complex disorders, including cancer, atherosclerosis, and cardiovascular- and autoimmune diseases [20,21]. Reliable biomarkers may help with the early diagnosis of disease and the implementation of a precision medicine treatment program. There are presently no reliable and accurate epigenetic biomarkers to aid the diagnosis and classification of diseases and for evaluating disease progression [22]. The majority of current biomarkers are based on costly and/or invasive procedures, such as biopsies, imaging, and cerebrospinal fluid analyses [22]. A liquid biopsy is a more comfortable yet less expensive option. Consequently, the search for epigenetic biomarkers has shifted towards using accessible fluids, such as blood. 5mC has recently been proposed as a novel biomarker for neurodegenerative and cardiovascular disorders [10,14].

The aim of the current study was to examine the effect of patient medications on global DNA methylation levels, specifically with respect to follow-up assessments in patients with hypovitaminosis, neuropsychiatric, cerebrovascular, and neurodegenerative disorders. These results were compared to other biochemical and psychometrical criteria to determine which patients respond better to treatment and have higher levels in global DNA methylation.

## 2. Materials and Methods

### 2.1. Subjects

For this retrospective study, patients (*n* = 98; age: 56 ± 1.61 years; range: 18–84 years) were recruited from the CIBE Database at the EuroEspes International Center of Neuroscience and Genomic Medicine (C000925, 21 October 2013, EuroEspes Biomedical Research Center, Bergondo, Spain). The study was carried out in accordance with the Helsinki Declaration, Spanish law (Organic Law on Biomedical Research, 14 July 2007), and with the approval of the Ethics/Research Committee of the EuroEspes Biomedical Research Center (Epibiomarkers EE0620). Blood samples were obtained from patients during the first consultation and then during the first follow-up visit (EuroEspes Biomedical Research Center). The samples were obtained after the informed consent of all patients and/or legal caregivers. Patients were diagnosed using globally accepted diagnostic criteria, following a comprehensive clinical and genetic examination. A genomic analysis of many SNPs connected to PD, AD, or vascular risk, as well as psychological tests, brain mapping, and neuroimaging were included in the clinical protocol for patients.

Patients were divided into two groups: those in Group A were patients with higher 5mC levels during the follow-up than during the first visit. Patients in Group B included those who had the same or lower 5mC levels during the follow up than during the initial visit. This classification was carried out manually, and only differences between 5mC levels greater than 0.2% allowed for the separation of patients into Groups A and B.

### 2.2. Sample Collection and Analysis

Venous blood samples were obtained from individuals in the supine position following overnight fasting. EDTA-coated tubes from peripheral blood were centrifuged at 3000 rpm at 4 °C for 10 min and the buffy coat was collected and stored at −40 °C until DNA extraction. Blood for the analysis of serum iron (Fe), folate, and vitamin B12 was collected in BD Vacutainer serum separation tubes. Samples in serum-containing tubes were allowed to clot for 30 min at room temperature [23]. The samples were centrifuged (3500 rpm) at 4 °C for 10 min, and the supernatant (serum) was then removed and stored at −80 °C.

Serum levels of vitamin B12 and folate were measured by electrochemiluminescence on the same day of venipuncture using the Cobas e-411 system (Roche Diagnostics, Mannheim, Germany) [24,25]. The normal range (for healthy individuals) as recommended in the commercial kit for serum folate is 3.89–26.8 ng/mL and for serum vitamin B12 it is 197–771 pg/mL.

Iron concentrations were measured by UV-visible spectrophotometry using a Cobas Mira Plus automated analyzer (Roche Diagnostics, Mannheim, Germany). The serum iron concentration was determined colorimetrically (Spinreact, Barcelona, Spain) according to the ferrozine method [23]. The reference concentration ranges used were 65–175 µg/dL for men and 40–150 µg/dL for women.

### 2.3. Neuropsychological Assessments

A standard series of neuropsychological tests were administered to subjects to differentiate between age-related deficits and cognitive deficits associated with dementia. The Mini-Mental State Examination (MMSE) was used as an initial screen for dementia [26]. The Alzheimer’s Disease Assessment Scale (ADAS) was used to assess cognitive function [27]. The ADAS test enables precise monitoring of disease and the assessment of therapeutic efficacy [27]. The Global Deterioration Scale (GDS) was used to measure the degree of cognitive decline [28]. The functional assessment staging (FAST), that examines the deterioration in daily functional activity, was used in conjunction with GDS. GDS/FAST differentiates seven phases of AD, ranging from normal aging (no deterioration in cognitive function) to severe dementia [29].

### 2.4. DNA Extraction

The QIAcube robotic workstation and QIAamp DNA Mini Kit (Qiagen, Hilden, Germany) were used to extract DNA from peripheral blood lymphocytes according to the manufacturer’s instructions. A microplate spectrophotometer (Epoch, BioTek Instruments, Winooski, VT, USA) was then used to measure DNA concentrations. DNA samples with 260/280 and 260/230 ratios above 1.8 only were used in this study.

### 2.5. Quantification of Global DNA Methylation (5mC)

Global 5mC levels were quantified colorimetrically with 50 ng DNA per sample using the MethylFlash Methylated DNA Quantification Kit (Epigentek, New York, NY, USA), as directed by the manufacturer. A microplate reader was then used to measure absorbance at 450 nm. We constructed a standard curve with linear regression (Microsoft Excel Version 18.2210.1203.0) to assess the absolute quantity of methylated DNA. The amount (ng) and percent of 5 mC was then calculated with the formula:5 mC (ng) = (Sample OD − Blank OD)/(Slope × 2)
5 mC (%) = 5 mC (ng)/sample DNA (ng) × 100

### 2.6. Statistical Analysis

The D’Agostini-Pearson and Levene’s tests were used to test for data normality and equality of variance, respectively. The statistical significance was determined with paired t tests or one-way ANOVA with a Bonferroni post-hoc analysis (GraphPad Prism, San Diego, CA, USA). Linear regression was used to perform a correlation analysis (GraphPad Prism). Data are presented as mean ± S.E.M.; * *p* < 0.05, ** *p* < 0.01 and *** *p* < 0.001 were considered statistically significant.

## 3. Results

### 3.1. Evaluation of Global DNA Methylation after the Initial Consult and after Treatment

Global DNA methylation is proposed as a novel biomarker for diagnosing a variety of complex diseases, including neurodegenerative disorders [16,30], vascular diseases [31,32], cancer [33], and other prevalent diseases. We first examined global DNA methylation levels in 98 patients who visited the EuroEspes International Center of Neuroscience and Genomic Medicine, during their initial visit and then during the follow-up. Patients received multifactorial treatment in the intervening period prior to clinical follow-up. In their first visit, all patients had 5mC levels of 2.98 ± 0.2%, which, during the follow-up visit, increased to 3.37 ± 0.27%, but this difference was not statistically significant (Figure 1A). However, the further analysis of 5mC levels revealed two distinct categories of patients: patients in Group A who, during the follow-up, exhibited an increase in 5mC levels (2.46 ± 0.15% to 3.95 ± 0.28%, *p* < 0.001) (Figure 1B), and Group B comprising of patients with decreased levels of 5mC (3.74 ± 0.32% to 2.46 ± 0.21%, *p* < 0.001 (Figure 1C).

### 3.2. Correlation Analysis between Global DNA Methylation and Iron, Folate, and Vitamin B12

To identify the reasons for the different responses in global DNA methylation between Groups A and B, next we analyzed iron, folate, and vitamin B12 levels in both patient groups. These biochemical parameters have been associated with DNA methylation and brain disorders [34,35,36]. We used linear regression to assess the relationship between 5mC and iron levels in Groups A and B (Figure 2A,B). In patient samples (*n* = 51) from Group A, there was no correlation between 5mC and iron levels (*p* = 0.944; r^2^ = 7.2 × 10^−005^) (Figure 2A). There was also no correlation between iron and 5mC levels (*p* = 0.1311; r^2^ = 0.0496) (Figure 2B) in samples from patients (*n* = 47) in Group B. Next, within each group of patients, we compared 5mC levels measured during the first visit with levels during the follow-up (Figure 2C,D). No statistically significant differences in iron levels were detected in either group between the first visit and the follow-up. Iron levels in samples from patients in Group A increased from 78.72 ± 2.8 µg/dL to 82.12 ± 1.23 µg/dL (Figure 2C) and decreased from 83.24 ± 4.01 µg/dL to 79.67 ± 1.56 µg/dL in patients from Group B (Figure 2D). These findings suggest, from the patient samples examined, that there is no correlation between iron and 5mC levels.

Folate is critical for generating S-adenosylmethionine (SAM), a key donor in reactions necessary for the methylation of DNA [37]. In the current study, there was a significant positive correlation (*p* = 0.0466) between folate and 5mC levels in patients (Group A) whose 5mC levels were higher during the follow-up visit than during the first consultation (Figure 3A). However, 5mC and folate levels were negatively correlated (*p* = 0.0272) in patients from Group B (Figure 3B). Compared to the first visit, folate levels were significantly higher in patients from Group A (*p* < 0.001) during the follow-up. However, a similar comparison found no statistically significant difference when analyzing folate levels in patients from Group B (Figure 3C,D). Interestingly, basal folate levels were higher in Group B (16.59 ± 2.56 ng/mL) than in Group A (13.26 ± 0.64 ng/mL). Folate levels in patients from Group A significantly increased from 13.26 ± 0.64 ng/mL to 18.79 ± 0.45 ng/mL (*p* < 0.001), but decreased from 16.59 ± 2.56 ng/mL to 13.49 ± 0.5 ng/mL in Group B patients (Figure 3C,D).

Other nutrients, such as vitamins B6 and B12, are required in addition to folate to sustain one-carbon flow, ensuring appropriate homocysteinase remethylation, SAM production, and DNA methylation. [36]. There was no correlation between vitamin B12 and 5mC levels in those patients (Group A) whose 5mC levels were higher during the follow-up than during the first visit (Figure 4A). There was, however, a non-significant negative correlation between vitamin B12 and 5mC levels in patients from Group B (Figure 4B). The levels of vitamin B12 were significantly higher in Group A patient samples during the follow-up (547.56 ± 27.75) than during the initial visit (459.17 ± 18.58) (Figure 4C). However, there were no differences in vitamin B12 levels between the initial (563 ± 23.9) and follow-up visits (579.17 ± 11.58) in patients from Group B (Figure 4D).

### 3.3. Correlation Analysis between Global DNA Methylation and Psychometric Assessments

Many of the patients who visit EuroEspes Biomedical Center are diagnosed with nervous system-related pathologies that include epilepsy, bipolar disorder, and neurodegenerative and cerebrovascular disorders. The MMSE is the best-known and most frequently used short screening tool to assess cognitive impairment and was conducted on all patients in the present study. When we analyzed patients from Group A and B together, there was a significant positive correlation between 5mC levels and MMSE scores (*p* = 0.0438) (Figure 5A). However, this correlation was more significant in patients (Group A) (*p* = 0.0110) with higher 5mC levels during the follow-up visit (Figure 5B) than in those patients who did not show an increase in 5mC levels during the second visit (Group B) (*p* = 0.8516) (Figure 5C).

We also analyzed the relationship between MMSE scores and folate levels (Appendix A), and between MMSE scores and vitamin B12 levels (Appendix A). There was no correlation between these parameters, not in pooled (patient Groups A and B) samples (Appendix A) and neither when we analyzed samples from patients in Group A only (Appendix A).

The ADAS scale comprises cognitive (ADAS-Cog) and non-cognitive (ADAS-Noncog) sections. The ADAS-Cog test assesses the main components of cognitive function and allows for the evaluation of the efficacy of pharmacologic interventions against dementia. ADAS is more sensitive than the MMSE [38]. In the current study, GDS staging was conducted on all patients over the ages of 50–55 years old. We found no correlation between 5mC levels and ADAS-Cog scores when we analyzed groups A and B together (Figure 6A). However, when we separated patients into groups based on whether their 5mC levels improved (measured during the initial visit and then during the follow-up), ADAS-Cog scores and 5mC levels were only negatively correlated in patients who showed an increase in 5mC levels (Group A) (Figure 6B,C). This finding, however, was statistically non-significant (*p* = 0.0785). When we analyzed the relationship between ADAS-Cog scores and folate levels, and between ADAS-Cog scores and vitamin B12 levels, there was only a statistically significant correlation between ADAS-Cog scores and vitamin B12 levels in the group of patients who showed improved 5mC levels due to treatment (Group A) (Appendix A).

Next, we used GDS staging to quantitatively assess the clinical course of functional decline in patients with neurocognitive deficits, mixed dementia, or cerebrovascular disorders. In the current study, GDS staging was conducted on all patients over the ages of 50–55 years old. A higher GDS score indicates greater cognitive impairment [28]. Patients with higher 5mC levels due to treatment had low GDS scores (stage 1, normal aging, 13.79%; and stages 1–2, normal aging–very low cognitive deficit, 13.79%) (Figure 7A). However, only 4.76% of patients who showed no increase in 5mC levels had a GDS score of 1 (Figure 7B). When we analyzed the percentage of patients with high GDS scores, we found the opposite: 6.89% and 3.44% of patients who showed an increase in 5mC levels had GDS scores of 4 (initial dementia) and 4–5 (initial–moderate dementia), respectively (Figure 7A). However, the percentage of patients who showed no improvement in 5mC levels increased to 14.29%, with GDS scores of 4 and 4–5 (Figure 7B).

### 3.4. Correlation Analysis between Global DNA Methylation and Clinical Diagnosis

Next, we examined patient diagnostic data (Table 1) to identify the disorders in which patients showed increased levels of 5mC (Group A, left) compared to those disorders in which patients did not exhibit elevated 5mC levels (Group B, right) after treatment. The percentage of patients with hypovitaminosis was 53.13% lower in patients in Group B than in Group A patients. Prior to follow-up, patients were prescribed conventional medications, as well as various vitamins (Becozyme C Forte, which contains vitamins B1, B2, B6, B7, B12, and C, nicotinamide, calcium pantothenate, and Hidroxil-B1-B6-B12), supplements (folate, 5 mg/day; Tardyferon/iron, 80 mg/day), and the nutraceutical bioproducts Animon Complex^®^ (two capsules/day), MineraXin Plus^TM^ (three capsules/day), NeoBrainine^®^ (two capsules/day), and AtreMorine^TM^ (5–20 g/day). Animon Complex contains a purified extract of *Chenopodium quinoa* (250 mg) enriched with folic acid (200 µg), vitamin B12 (2.5 µg), and ferrous sulphate [39] (14 mg). MineraXin Plus, an extract (250 mg) derived from the Mediterranean mussel *Mytilus galloprovincialis*, contains natural mono- and polyunsaturated fatty acids (mainly of the omega 3-type), choline, ceramides and phosphatidylethanolamines, vitamins A, B9, B12, C, D3, and E, and minerals (iodine, iron, selenium, sodium, phosphorus, and zinc) [40]. NeoBrainine is a potent nootropic agent that contains citicoline (cytidine diphosphate-choline or CDP-choline, 500 mg), niacine (niacinamide) (16 mg) and calcium D-pantothenate (6 mg) and is used for the promotion of brain health [41]. In the current study, NeoBrainine was used to treat patients with neurocognitive deficits, dementia, and cerebrovascular disorder. AtreMorine, an extract derived from the *Vicia faba* L. plant, is a natural source of 3,4-dihydroxy-L-phenylalanine (L-DOPA, 20 mg/g of AtreMorine) and is a potent dopamine enhancer used to treat patients with PD [42,43]. The incidence of patients with PD was higher in Group A (20.69%) than in Group B (6.25%), while the incidence of anxiety/depression was slightly lower in patients in Group A (27.59%) than in Group B (26.67%). Vascular headache was only diagnosed in patients in Group A (6.90%).

### 3.5. Analysis of the Association between Global DNA Methylation and Pharmacotherapy

The patients in our study not only received a multifactorial treatment regimen that included vitamins, supplements, and nutraceutical bioproducts, but also conventional treatments for neurological disorders. Patients with vascular headache received Actron (Ketoprofen) and/or Tryptizol (Amitriptyline) (10–50 mg/day); patients with moderate-to-severe depression received Tryptizol (Amitriptyline) (10–50 mg/day) and/or Deprax (Trazodone) (100 mg/day); patients with moderate-to-severe anxiety and/or agitation received Diazepam (2.5–5 mg/day); patients with cerebrovascular disorders received Varson (Nicergoline) (5 mg/day); and patients with psychotic disorders received Somazine (Citicoline) (500 mg/day). Since 80% of patients in Group A (that is, those subjects with increased 5mC levels) exhibited hypovitaminosis (Table 1), we investigated whether there were differences in the levels of global DNA methylation in those patients between the initial- and follow-up visits in the presence of and absence of treatment with folate, Hidroxil-B1-B6-B12, Tardyferon, Animon Complex, and MineraXin Plus. Patients treated with folate (*p* = 0.011) (Figure 8A), Hidroxil-B1-B6-B12 (*p* = 0.034) (Figure 8B), Animon Complex (*p* = 0.0058) (Figure 8D), and MineraXin Plus (*p* = 0.0038) (Figure 8E) showed significant increases in 5mC levels during the follow-up compared to the initial visit. There were no differences in 5mC levels between the initial- and follow-up visits in patients from Group A in response to treatment with Tardyferon (Figure 8C).

Many patients in Group A were diagnosed with neuropsychiatric, cerebrovascular, and neurodegenerative disorders. Therefore, we next analyzed whether there were differences in the levels of global DNA methylation in those patients with neurological disorders between the initial- and follow-up visits in the presence of and absence of treatment with conventional drugs (Actron, Tryptizol, Deprax, Diazepam, Varson, and Somazine) and with nutraceutical bioproducts (AtreMorine and NeoBrainine). Patients who were or were not treated with Actron exhibited higher 5mC values during the follow-up- (*p* = 0.01) than during the initial visits (*p* = 0.014) (Figure 9A). However, in patients who were not treated with Tryptizol (*p* < 0.001) (Figure 9B), Deprax (*p* = 0.0049) (Figure 9C), Diazepam (*p* = 0.0069) (Figure 9D), Varson (*p* = 0.009) (Figure 9E), and Somazine (*p* = 0.031) (Figure 9F), there were significant increases in 5mC levels during the follow-up compared to the initial visit. This relationship between 5mC levels and pharmacotherapy was not observed during follow-up in patients who were treated with these drugs.

Patients not treated with AtreMorine showed significantly higher 5mC levels during the follow-up- (*p* = 0.014) than during the initial visit (Figure 9G). Significantly higher 5mC levels were also detected during the follow-up than during the initial visit in patients who were treated with AtreMorine (*p* = 0.048). Similarly, patients not treated with NeoBrainine exhibited significantly higher 5mC levels during the follow-up- (*p* = 0.028) than during the initial visit (Figure 9G). 5mC levels were also significantly higher during the follow-up than during the initial visit in patients who were treated with NeoBrainine (*p* = 0.021) (Figure 9H).

## 4. Discussion

In this study, we showed that there were two groups of patients, one in which 5mC levels improved during the clinical follow-up (Group A), and another (Group B) in which 5mC levels were lower during the follow-up, than during the initial visit. These groups included patients with hypovitaminosis, neuropsychiatric (anxiety, bipolar, and psychotic disorders, depression), cerebrovascular, neurodegenerative disorders (PD, mixed dementia), and other nervous system-related pathologies (neurocognitive deficits, epilepsy, and vascular headache). Our goal was to investigate this discrepancy in patient response between each of these two groups, and to identify the parameters associated with these different responses that may be useful for monitoring the treatment outcome.

The dysregulation of DNA methylation has been linked to a number of brain disorders, such as AD, schizophrenia, and autism. DNA methylation has been proposed as an epigenetic biomarker for neurodegenerative disorders [10,14,44]. AD, for example, is associated with global changes in DNA methylation, which may be a useful indicator for the clinical assessment of neurodegenerative diseases. In the current study, the analysis of 5mC levels (initial vs. follow-up) in serum from our patient cohort revealed two categories of patients; Group A who, during the follow-up, had increased 5mC levels, and Group B comprising patients with decreased levels of 5mC. We furthermore examined the relationships between Fe-, folate-, vitamin B12- levels, and 5mC levels in these subjects. Patients with low Fe-, folate-, and vitamin B12- levels during the initial visit showed increased levels of 5mC in response to treatment when assessed during the follow-up. The analysis of the relationship between global DNA methylation and psychometric assessments indicated a positive correlation between 5mC levels and MMSE scores, and an inverse correlation between 5mC and ADAS-Cog scores, but only in Group A patients. GDS staging showed that patients with higher 5mC levels during the follow-up had low GDS scores. There is a positive correlation between global DNA methylation levels and age [17]. However, our group previously found a significant correlation between age and 5mC levels only in patients with PD [10]. In the present study, we found no correlation between age and 5mC levels in Group A patients (Appendix A).

Maintaining normal folate (vitamin B9) levels helps prevent aberrant DNA methylation and ensures correct transcriptional regulation. Gene polymorphisms, poor diet, chronic smoking and alcoholism, gastrointestinal diseases, and medications all contribute to chronic folate deficiency [45]. Moreover, low folate levels are associated with a variety of diseases, including cancer and cardiovascular- and neurodegenerative disorders [35]. There is an association between folate deficiency and high homocysteine levels in AD, PD, and stroke [46]. Low levels of folic acid in rodents during weaning causes alterations in gene expression associated with DMNTs in the brain, and a reduction in the number of proliferating cells in the hippocampus in neonatal and adult mice [43]. In patients with AD, methylation in ten CpG sites in the *SNCA* gene that encodes α-synuclein (SNCA) is decreased [47]. Furthermore, many patients with dementia consistently have decreased plasma folate levels [48]. In AD, following folate deficiency-induced hypomethylation, the pathogenic genes β-site amyloid precursor protein cleaving enzyme *(BACE)-1* and presenilin 1 (*PS1*) show increased expression, and this effect is reversed with the dietary supplementation of SAM [49]. Chen et al. further showed that supplementation with folic acid plus vitamin B12 has an beneficial effect on ADAS scores [50]. In the current study, folate therapy in Group A patients between the initial and follow-up clinical visits may therefore explain the increase in their levels of 5mC. Since DNA methylation patterns are reversible, they may therefore be used as a sensitive marker for monitoring disease progression, evaluating treatment response and possibly predicting the therapeutic outcome.

There is a strong correlation between one-carbon metabolism nutrients, such as vitamin B12, folic acid, and DNA methylation [51]. Vitamin B12 deficiency causes DNA hypomethylation in the TCblR/CD320-knockout mouse brain [52]. Furthermore, in patients with colorectal cancer, there is an association between the levels of global DNA methylation and serum vitamin B12 [53]. Vitamin B12 therapy between the first-and follow-up visits may explain the improvement in 5mC levels in serum samples from patients in Group A. Interestingly, the incidence of hypovitaminosis was lower in those patients with the poorest evolution in 5mC levels. Patients with lower folate and/or vitamin B12 levels showed higher 5mC levels during the clinical follow-up after dietary intervention with vitamins and the nutraceutical bioproducts Animon Complex and MineraXin Plus. This finding emphasizes the need for adequate vitamin levels to ensure the proper functioning of the epigenetic machinery. Given the roles of vitamins B9 and B12 in DNA methylation, the improvement in patients in Group A that were diagnosed with hypovitaminosis may explain the increase in 5mC levels in those individuals.

There was a significant correlation between MMSE scores and 5mC levels only in patients (Group A) (*p* = 0.0110) that had higher 5mC levels during the follow-up than during the initial consult. However, there was no correlation between MMSE scores and folate levels, or between MMSE scores and vitamin B12 levels in pooled samples, or in samples from patients in Group A. ADAS-Cog scores inversely correlated with 5mC levels only in Group A patients who showed an increase in 5mC levels. Furthermore, there was a significant correlation between ADAS-Cog scores and vitamin B12 levels in patients (Group A) who had elevated 5mC levels due to treatment. Furthermore, Chen et al. showed that supplementing the diet of AD patients with vitamin B12, as well as folic acid, improves ADAS scores [50].

In terms of the relationship between global DNA methylation and patient diagnostics, there were substantially higher percentages of patients with PD and hypovitaminosis in Group A (correlating to higher 5mC levels) than in Group B. There was an association between the MMSE and GDS tests and global DNA methylation during the follow-up; patients with low scores in both tests had low levels of, and showed the slowest improvement in, 5mC levels. Differential methylation in genes associated with cognitive impairment, cognitive decline, Wnt signaling, and mitochondrial apoptosis is associated with cognitive and motor progression in PD [54]. For example, according to the MMSE administered during the follow-up visit in PD patients, methylation levels at CpG sites in the *KCNB1*, *DLEU2,* and *SATB1* genes are associated with faster cognitive decline and loss of punctuation [54]. Patients with AD who perform well on the MMSE exhibit higher levels of long interspersed element-1 (LINE-1) methylation than those who perform poorly [55]. These findings corroborate the data in the current study, showing that patients with higher MMSE scores had higher levels of global DNA methylation.

As previously stated, Group A had a higher percentage of patients with hypovitaminosis than patients in Group B. We found during the follow-up, that treatment with different vitamins, dietary supplements, or nutraceutical bioproducts increased 5mC levels in patients in Group A. During the follow-up, patients treated with the nutraceutical bioproducts Animon Complex and MineraXin Plus showed higher levels of DNA methylation compared to patients treated with folate, Hidroxil-B1-B6-B12, or Tardyferon. Animon Complex and MineraXin Plus are produced by non-denaturing biotechnological processes that enable the preservation of the healthy properties of the original species. Since Animon Complex and MineraXin Plus contain multiple pharmacologically and biologically active ingredients, including vitamins and minerals, this may explain why these nutraceutical bioproducts produce stronger effects on global DNA methylation than the administration of single agents.

In the current study, we observed a general reduction in 5mC levels during follow-up in patients treated with conventional medications (e.g., Deprax and Diaze-pam) for several types of neurological disorders. It is possible that these drugs interact with the DNA methylation machinery, inhibiting this epigenetic process. However, the treatment of patients with the nutraceutical bioproducts AtreMorine or NeoBrainine maintained an increase in 5mC levels during the follow-up compared to the initial consultation. This suggests that these compounds preserve and maintain the functional activity of the DNA methylation machinery and act as epidrugs. Previously, we have described that AtreMorine regulates DNA methylation in AD and PD [56]. Citicoline, a key component in Neobrainine, is a source of choline. Choline is an essential nutrient that participates in the synthesis of S-adenosyl methionine, a major methyl donor for histone methyltransferases, and is therefore important in DNA methylation [57]. Moreover, citicoline increases the levels of acetylcholine, dopamine, and norepinephrine, promotes the integrity of neuronal cell membranes, and enhances ATP generation in the frontal cortex. In the present study, the majority of patients with brain disorders received a combination of conventional and non-traditional treatments. These data appear to indicate that AtreMorine and NeoBrainine protect the DNA machinery against the effects of conventional drugs on 5mC levels in patients with neurological disorders. Since AtreMorine, NeoBrainine, Animon Complex, and MineraXin Plus regulate the epigenome by targeting epigenetic marks, they may therefore be considered epidrugs.

## 5. Conclusions

Global DNA methylation is reduced in different pathologies and 5mC has therefore been proposed as a diagnostic biomarker. Given that epigenetic changes are reversible, global DNA methylation may be utilized clinically as a biomarker to evaluate treatment outcome. In this study, only a subset of patients in our cohort showed higher 5mC levels in response to treatment. Prior to therapy, this group of patients had lower folic acid and vitamin B12 levels with a higher prevalence for the diagnosis of hypovitaminosis. After treatment with vitamin supplements, 5mC levels increased in those patients (Group A), and the percentage of patients in that same group who had been diagnosed with hypovitaminosis decreased. There was a correlation between MMSE scores and 5mC levels, particularly in those patients with increased 5mC levels in response to treatment. Moreover, low GDS scores were more prevalent in patients who had responded successfully to treatment and had higher 5mC levels during the clinical follow-up. In conclusion, patients with hypovitaminosis and/or with better neuropsychometric scores had higher 5mC values following treatment, suggesting that global DNA methylation has diagnostic importance as a biomarker across different pathologies.

## Figures and Tables

**Figure 1 genes-14-00365-f001:**
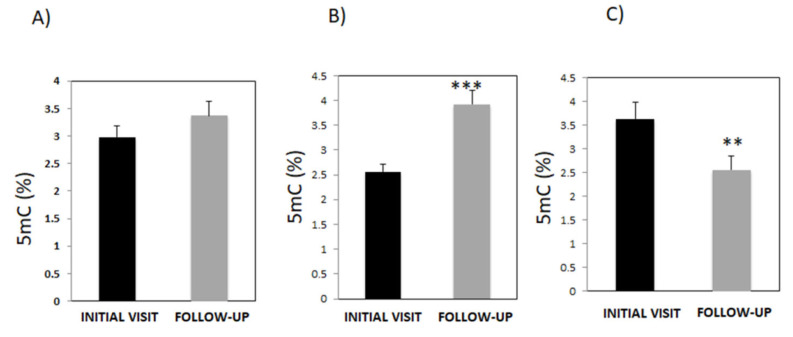
Analysis of global DNA methylation (5mC, %) in blood samples from patients with the diagnoses of neurological disorders and/or hypovitaminosis. (**A**) 5mC levels were measured in these patients (*n* = 98) during the first visit and then during the clinical follow-up. (**B**) Patients (*n* = 51; Group A) whose 5mC levels were found to be higher during the follow-up visit than during the initial consult. (**C**) Patients (*n* = 47; Group B) whose 5mC levels decreased during the follow-up visit versus the initial visit. Data are presented as the mean ± S.E.M; paired *t* test (** *p* < 0.01, *** *p* < 0.001). 5mC, 5-methylcytosine.

**Figure 2 genes-14-00365-f002:**
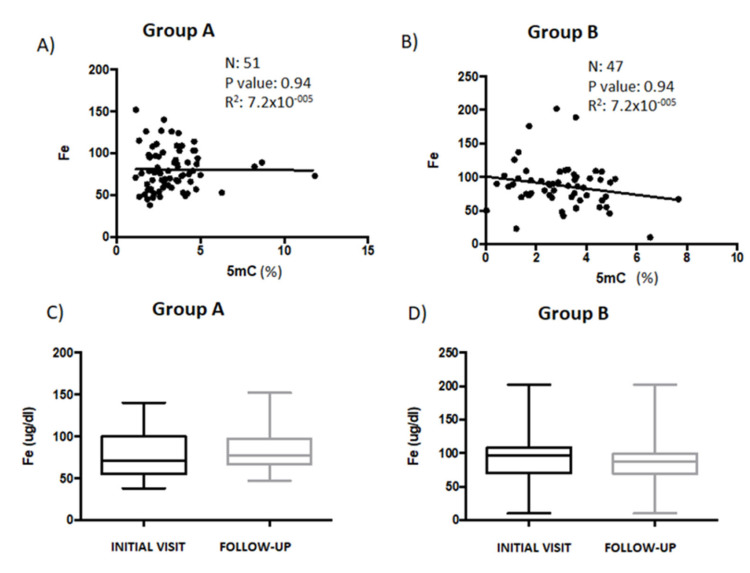
The relationship between Fe levels and global DNA methylation in patient serum samples. (**A**) Correlation analysis to test the association between Fe and 5mC levels in serum samples from patients in Group A (*n* = 51) and (**B**) from patients in Group B (*n* = 47). Fe levels measured during the initial- and follow-up visits were compared between patients in (**C**) Group A and (**D**) Group B. Data are presented as the mean ± S.E.M; paired *t* test. 5mC, 5-methylcytosine; Fe, iron.

**Figure 3 genes-14-00365-f003:**
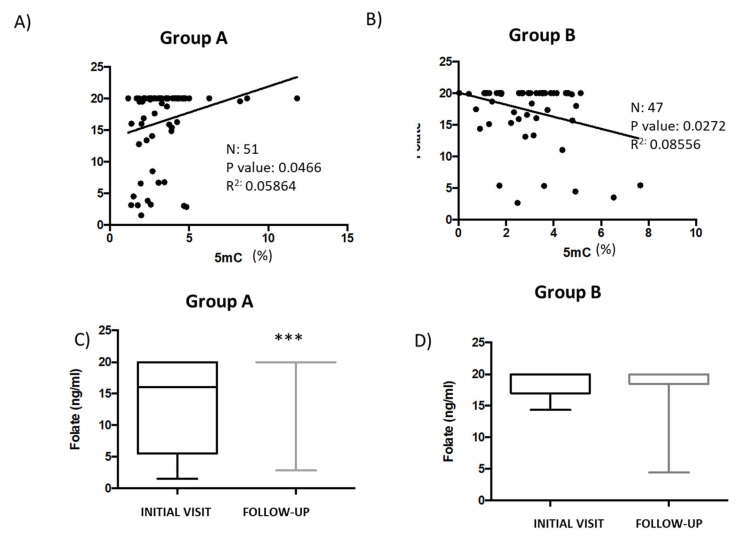
The relationship between folate levels and global DNA methylation in patient serum samples. (**A**) Correlation analysis to test the association between folic acid and 5mC levels in serum samples from patients in Group A (*n* = 51) and (**B**) from patients in Group B (*n* = 47). Data points indicate individual patient values. Folate levels measured during the initial- and follow-up consults were compared between patients in (**C**) Group A and (**D**) Group B. Data are presented as the mean ± S.E.M; paired *t* tests (*** *p* < 0.001). 5mC, 5-methylcytosine.

**Figure 4 genes-14-00365-f004:**
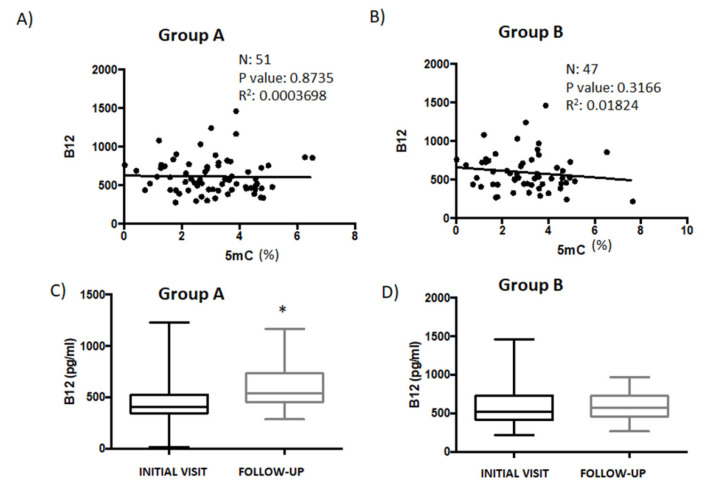
The relationship between vitamin B12 level and global DNA methylation in patient serum samples. (**A**) Correlation analysis to test the association between vitamin B12 and 5mC levels in serum samples from patients in Group A (*n* = 51) and (**B**) from patients in Group B (*n* = 47). Vitamin B12 levels measured during the initial- and follow-up consults were compared between patients in (**C**) Group A and (**D**) Group B. Data are presented as the mean ± S.E.M; paired *t* test (* *p* < 0.05). 5mC, 5-methylcytosine.

**Figure 5 genes-14-00365-f005:**
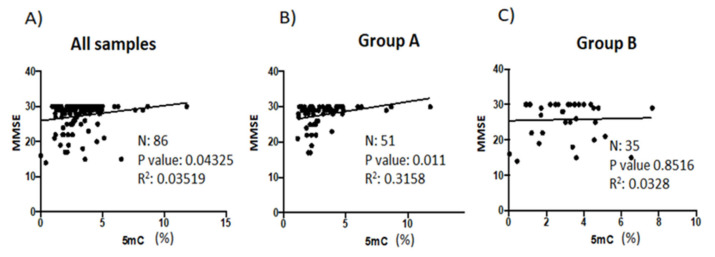
Correlation analyses of the relationship between patient MMSE scores and levels of global DNA methylation during clinical follow-up. (**A**) The association between MMSE scores and 5mC levels in (**A**) all patients (*n* = 86), (**B**) patients in Group A (*n* = 51), and (**C**) patients in Group B (*n* = 35). Data points indicate individual patient values. Pearson correlation was used to establish statistical correlation. 5mC, 5-methylcytosine; MMSE, Basal Mini-Mental State Examination.

**Figure 6 genes-14-00365-f006:**
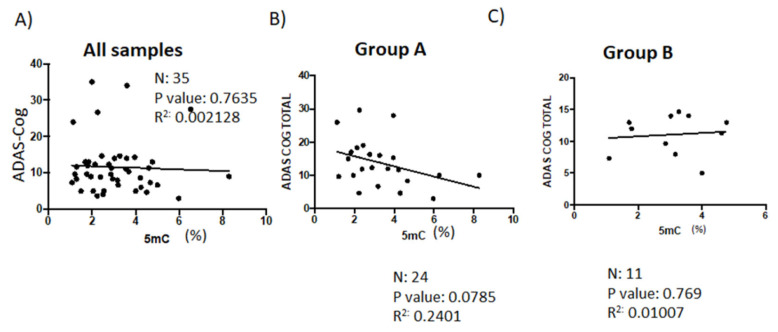
Correlation analyses of the relationship between patient ADAS-Cog score and levels of global DNA methylation during clinical follow-up. (**A**) The association between the total ADAS-Cog scores and 5mC levels in all patients (*n* = 35). (**B**) Correlation analysis to test the association between total ADAS-Cog scores and 5mC levels in patients from Group A (*n* =24), and (**C**) patients from Group B (*n* =11). Data points indicate individual patient values. Pearson correlation was used to establish statistical correlation. 5mC, 5-methylcytosine; ADAS, Alzheimer’s Disease Assessment Scale; ADAS-Cog, ADAS-cognitive.

**Figure 7 genes-14-00365-f007:**
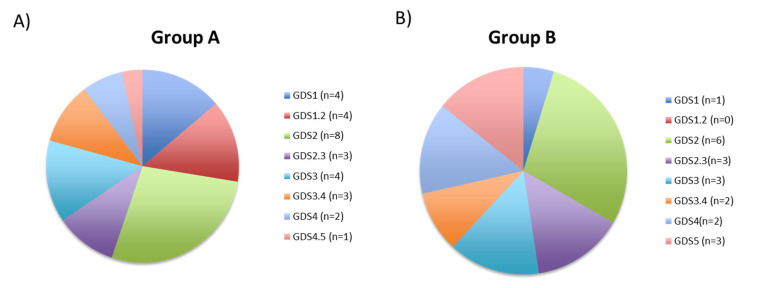
Pie charts depicting the relationship between GDS stages and levels of global DNA methylation (5mC) in (**A**) patients (*n* = 29; Group A) whose 5mC levels were found to be higher during the follow-up consultation than during the initial visit, and (**B**) patients (*n* = 21; Group B) whose 5mC levels did not increase during the follow-up versus the initial visit. GDS stages 1 and 2 indicate normal cognitive function, and higher GDS stages indicate progressive cognitive impairment. 5mC, 5-methylcytosine; GDS, Global Deterioration Scale.

**Figure 8 genes-14-00365-f008:**
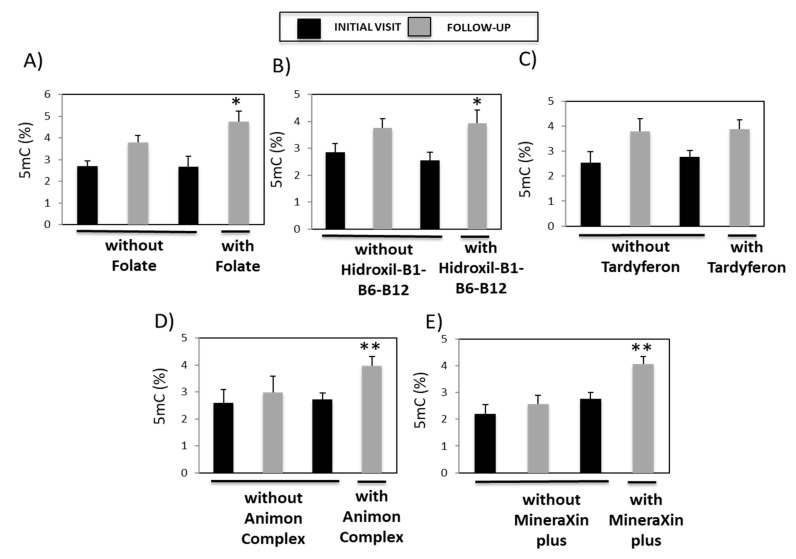
The relationship between vitamins, iron supplements, and nutraceutical bioproducts and global DNA methylation in patient serum samples. 5mC levels were compared in patients whose 5mC levels were higher during the follow-up than during the initial visit and who were, or were not, treated with (**A**) folate, (**B**) Hidroxil-B1-B6-B12, (**C**) Tardyferon, (**D**) Animon Complex^®^, and (**E**) MineraXin Plus^TM^. Data are presented as the mean ± S.E.M; one way ANOVA and Bonferroni post hoc test (* *p* < 0.05; ** *p* < 0.01). 5mC, 5-methylcytosine.

**Figure 9 genes-14-00365-f009:**
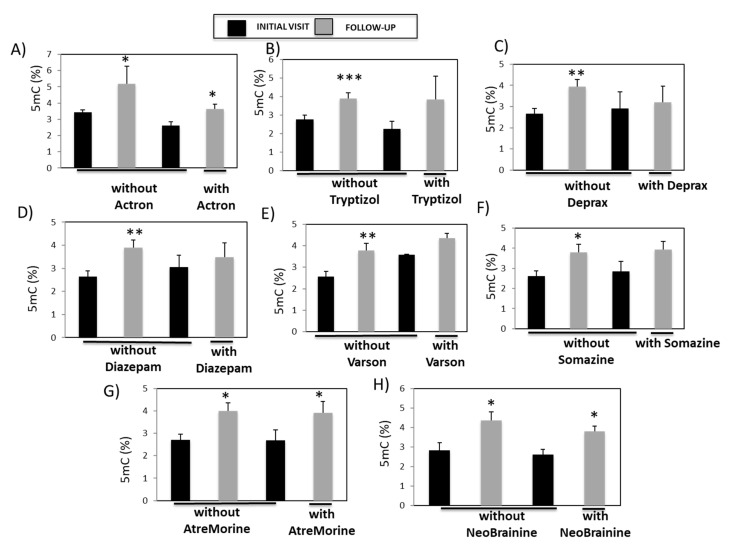
The relationship between pharmacotherapies for brain disorders and global DNA methylation in patient serum samples. Global DNA methylation levels were compared in patients whose 5mC levels were higher during the follow-up than during the initial visit and who were, or were not, treated with (**A**) Actron, (**B**) Tryptizol, (**C**) Deprax, (**D**) Diazepam, (**E**) Varson, (**F**) Somazine, (**G**) AtreMorine^TM^, and (**H**) NeoBrainine^®^. Data are presented as the mean ± S.E.M; one way ANOVA and Bonferroni post hoc test (* *p* < 0.05; ** *p* < 0.01; *** *p* < 0.001). 5mC, 5-methylcytosine.

**Table 1 genes-14-00365-t001:** Clinical diagnosis. Distribution of patient diagnostics indicating the percentage of patients with (A) increased 5mC levels (Group A) and (B) unchanged levels of 5mC (Group B) in serum during the clinical follow-up.

GROUP A		GROUP B	
Clinical diagnosis	%	Clinical diagnosis	%
Hypovitaminosis	80	Hypovitaminosis	37.5
Neurocognitive deficit	13.79	Neurocognitive deficit	18.75
Parkinson’s disease	20.69	Parkinson’s disease	6.25
Mixed dementia	13.79	Mixed dementia	18.75
Vascular headache	6.90	Cerebrovascular disorder	18.75
Cerebrovascular disorder	24.14	Anxiety/Depression	26.67
Anxiety/Depression	27.59	Others (Bipolar disorder, Epilepsy, Psychotic disorder)	25
Others (Bipolar disorder, Epilepsy, Psychotic disorder)	17.24		

## Data Availability

Not applicable.

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
