# Peer review of "DNA Methylation as a Biomarker for Monitoring Disease Outcome in Patients with Hypovitaminosis and Neurological Disorders"

_genes, 2023, doi:10.3390/genes14020365_

Round 1

Reviewer 1 Report

In this manuscript, the authors examined global DNA methylation (5mC) levels in serum samples from patients during the initial and follow-up visits. They found a group of patients, who received multifactorial treatment, had increased 5mC levels in the follow-up visits. They also presented correlational analyses between 5mC levels and vitamin levels, psychometric assessment scores, clinical diagnosis, and pharmacotherapy. The data presented in this manuscript may be beneficial to the community in the field. However, there are several issues in the manuscript that should be further addressed by the authors:

Major comments:

1. The definition of group A and group B is the basis of all the analyses in the manuscript, but the authors didn't clarify how they separated the patients into these two groups. Was it just manually selected by the authors, or was it dependent on any statistical test of the increases/decreases of the 5mC levels? These should be described in the method section and also in the main text.

2. Bar plots with error bars can be deceptive. For example, in Fig.2A, it's clear that there are some extreme values of 5mC around 10% in group A, which can not be seen in those bar plots. I would recommend showing the real distribution using violin plots or box plots, with the raw values of each data point overlaid on top. It will be even better if the authors can provide raw data in the form of supplementary tables.

3. In the method section, the authors said they used unpaired t-tests. The data were collected from the same patients, in the initial and follow-up visits. Paired tests seem to be more appropriate in this case. The authors should elaborate on why they made this decision in the analysis.

Minor issues:

1. In lines 171-174, the number of samples shown in Group A (n=70) and Group B (n=57) didn't match with those in Fig.1 (n=51 in Group A and n=47 in Group B).

2. In lines 197-200, the basal folate levels in Group B didn't match (15.52 ± 0.63 ng/ml and 16.59 ± 2.56 ng/ml).

3. In Fig.3A-B, why did the Folate levels cap at 20ng/ml? In the method (line 104), it said "The reference ranges supplied with the kits for serum folate was 3.89–26.8 ng/mL". Is this the reason (the numbers don't match though)?

4. In Fig.7, please label the exact numbers of each group in the pie chart.

Author Response

REVIEWER 1

Major Comment 1:

The definition of group A and group B is the basis of all the analyses in the manuscript, but the authors didn't clarify how they separated the patients into these two groups. Was it just manually selected by the authors, or was it dependent on any statistical test of the increases/decreases of the 5mC levels? These should be described in the method section and also in the main text.

Response:

Thank you to the reviewer. Patients were divided into two groups: those in Group A were patients with higher 5mC levels during the follow-up than during the first visit. Patients in Group B included those who had the same or lower 5mC levels during the follow up than during the initial visit. This classification was done manually, and only differences between 5mC levels greater than 0.2% allowed for the separation of patients into Groups A and B. This information has been added to the Materials and Methods section (paragraph 2) in the revised manuscript.

Major Comment 2:

Bar plots with error bars can be deceptive. For example, in Fig.2A, it's clear that there are some extreme values of 5mC around 10% in group A, which cannot be seen in those bar plots. I would recommend showing the real distribution using violin plots or box plots, with the raw values of each data point overlaid on top. It will be even better if the authors can provide raw data in the form of supplementary tables.

Response:

In Figures 2, 3, and 4, bar graphs have now been replaced with box plots.

Major Comment 3:

In the method section, the authors said they used unpaired t-tests. The data were collected from the same patients, in the initial and follow-up visits. Paired tests seem to be more appropriate in this case. The authors should elaborate on why they made this decision in the analysis.

Response:

Thank you to the reviewer for pointing this out. The authors apologize for this error. We have performed paired t tests, and this information has now been updated under “Statistical analysis” in the Materials and Methods section, and in the Figure legends.

Minor Comment 1:

In lines 171-174, the number of samples shown in Group A (n=70) and Group B (n=57) didn't match with those in Fig.1 (n=51 in Group A and n=47 in Group B).

Response:

This error has been rectified in the revised manuscript. The correct numbers are: n = 51 in Group A and n = 47 in Group B.

Minor Comment 2:

In lines 197-200, the basal folate levels in Group B didn't match (15.52 ± 0.63 ng/ml and 16.59 ± 2.56 ng/ml).

Response:

Thank you to the reviewer. We have corrected this error in the revised manuscript. The correct value for the basal folate levels in Group B patients is 16.59 ± 2.56 ng/mL.

Minor Comment 3:

In Fig.3A-B, why did the Folate levels cap at 20ng/ml? In the method (line 104), it said "The reference ranges supplied with the kits for serum folate was 3.89–26.8 ng/mL". Is this the reason (the numbers don't match though)?

Response:

The "reference ranges" refer to the normal range in healthy people, and not the range of detection. We have followed the reviewer’s suggestion and clarified this concept in the text.

Minor Comment 4:

In Fig.7, please label the exact numbers of each group in the pie chart.

Response:

Following the reviewer's recommendation, the n numbers of GDS-staged patients in each group have now been included in the pie chart in Figure 7.

Reviewer 2 Report

Martinez-Iglesias et al. investigated the changes in global DNA methylation (5mC) levels in serum samples from patients during the initial- and follow-up visits. In this retrospective study, the authors examined several factors to determine the possible reason to explain the increased global DNA methylation level in a group of patients during the follow-up (Group A patients). Although this study examined several factors that may explain the global DNA methylation level, there is still a concern about other possible confounding factors associated with global DNA methylation level changes.

(1) Previous studies found that the global DNA methylation level correlates well with age. Thus, can the authors include “age” as a confounding factor in the single variable statistical analysis that tests the association between other possible factors and two groups of patients?

 (2) Most analyses in this study are “single variable tests.” Can the authors also try multi-variable analysis to show the importance of each factor in the global DNA methylation level changes?

Author Response

REVIEWER 2

Comment 1:

Previous studies found that the global DNA methylation level correlates well with age. Thus, can the authors include “age” as a confounding factor in the single variable statistical analysis that tests the association between other possible factors and two groups of patients?

Response:

Thank you to the reviewer. Previous studies have shown that global DNA methylation levels correlate with age [1]. However, our group previously found a significant correlation between age and 5mC levels only in patients with Parkinson’s disease [2]. In the current study, we did not find any correlation between age and 5mC levels in patients in Group A. This information has been included in the Discussion section in the revised manuscript. In our resubmitted manuscript, we have also added a Supplementary Figure 1 that shows the correlation between age and 5mC levels in Group A patients.

References:

  1. Salemeh Y.; Bejaoui Y.; El Hajj N. DNA methylation biomarkers in aging and age-related disease. Front Genetics 2020, 11, 171.
  2. Martínez-Iglesias O.; Carrera I.; Carril J.C.; Férnandez_Novoa L.; Cacabelos N.; Cacabelos R. DNA methylation in neurodegenerative and cerebrovascular disorders. Int J Mol Sci 2020, 21, 2220.

Comment 2:

Most analyses in this study are “single variable tests.” Can the authors also try multi-variable analysis to show the importance of each factor in the global DNA methylation level changes?

Response:

Thank you to the reviewer for this helpful comment. Following the recommendations of the reviewer, we performed multiple regression analysis using MedCalc software. Unfortunately, when we analyzed the relationship between folate, iron, and vitamin B12 using multivariable analysis, there were no significant differences. However, as described in the manuscript, we did find differences in folate and B12 levels between patients in Groups A and B but only when we used single-variable tests (paired t tests). Although a single correlation analysis of the relationship between 5mC and folate revealed a significant p value (0.0466), this result was close to being statistically non-significant. Indeed, there was no significant univariable correlation between 5mC levels and iron or vitamin B12.